# CONSTRUCTING INFORMATIVE SUBTASK REPRESENTATIONS FOR MULTI-AGENT COORDINATION

## ABSTRACT

The introduction of subtasks holds the promise of promoting coordination in scenarios without communication. Instead of manually defined subtasks, recent studies attempt to decompose the overall task and allocate subtasks to agents automatically, but it remains unclear how to acquire a set of proficient subtask representations. In essence, the subtasks serve as auxiliary signals that assist agents in deducing the broader context from limited observations. To embed maximal information into subtask representations, we propose to first learn a vector quantised-variational autoencoder which takes individual observations of agents as inputs and reconstructs the global state based on their assigned subtasks as latent variables. Next, the informative representations can be readily integrated into various classic multi-agent reinforcement learning frameworks to facilitate insightful decisions of agents. Experiments on StarCraft II micro-war challenges and Google Research Football have demonstrated that our method learns reasonable and informative subtask representations, which facilitate the decision-making of agents and significantly improve the overall performance.

## 1 INTRODUCTION

Recently, multi-agent reinforcement learning (MARL) has been regarded as a powerful tool for solving complex cooperative tasks, such as MOBA games (Vinyals et al., 2019; Berner et al., 2019; Ye et al., 2020), flocking control (Pham et al., 2018; Xu et al., 2018), autonomous driving (Shamsoshoara et al., 2019), and traffic control (Chu et al., 2019; Wu et al., 2020). In multi-agent tasks, each agent faces a non-stationary environment due to the updating policies of other agents during training. Besides, the joint observation and action of all agents generate an extensive solution space which poses significant challenges in finding the optimal solution. These problems severely impede the performance of MARL algorithms.

Most MARL methods follow the *centralized training and decentralized execution* (CTDE) (Lowe et al., 2017) paradigm, which utilizes global information to train decentralized policies for all individuals. To address large-scale problems, recent approaches, including value-based (Sunehag et al., 2017; Rashid et al., 2018; Son et al., 2019; Yang et al., 2020; Wang et al., 2020a; Rashid et al., 2020) and policy-based(Lowe et al., 2017; Foerster et al., 2018), commonly share network parameters between agents. This technique markedly diminishes the quantity of trainable parameters and reduces the complexity of the model. Additionally, it allows agents to leverage each other's experience through concurrent parameter updates, resulting in mutual benefits. The aforementioned advantages facilitate the enhancement of training efficiency, but they also result in new challenges. Agents with shared parameters often exhibit similar behaviors (Li et al., 2021), thereby suggesting that the joint policy is stuck in a local optimum and unable to realize sophisticated coordination. For example, workers in Ford's factory undertake different tasks to produce automobiles collaboratively, and the required skills can hardly be represented by a single set of parameters Wang et al. (2020b).

To reach a balance between training efficiency and diversity of individual policies, the concept of *subtask* (or *role*) is introduced into MARL. In particular, agents are categorized into distinct groups, where agents within the same group focus on the same subtask, thereby learning a shared policy. The key question is how to decompose the task into a set of subtasks. Previous works often rely on subtasks manually defined by experts (Becht et al., 1999; Stone & Veloso, 1999; Pavón & Gómez-Sanz, 2003; Spanoudakis & Moraitis, 2010; Lhaksmana et al., 2018), which prevents

such approaches from extending to complex tasks without prior domain knowledge. Several recent studies seek to automatically encode the subtasks. ROMA (Wang et al., 2020b) learns a subtask encoder that maximizes the mutual information (MI) between the agent's subtask and trajectory. LDSA (Yang et al., 2022) maps the one-hot vectors of the subtasks to distinct high-dimensional representations. Nevertheless, these approaches aim to build differentiated subtask representations but neglect the information that ought to be incorporated within the subtask. Ideally, an agent can attain partial awareness of the global situation and adjust its behavior according to its assigned subtask. RODE (Wang et al., 2020c) suggests that only a subset of actions is needed in a certain subtask. It first learns the representation of each agent based on its effect on the environment then clusters the actions into subtasks. RODE establishes a new state-of-the-art (SOTA), but it may fail to address particular subtasks due to the restricted action space.

We believe that a proficient subtask representation should reflect the global information to a certain extent, thereby aiding the agent's decision-making. In this paper, we propose to Learn Subtask representations via Vector Quantization (LSVQ) for cooperative multi-agent reinforcement learning, rooted in the notion of subtasks. We first design a novel subtask learner with a structure akin to a variational autoencoder that attempts to reconstruct the global state through discrete subtask representations. Then, we incorporate the subtask learner into classic MARL frameworks to allocate subtasks to each agent. The hypernetwork (Ha et al., 2016) is adopted to generate subtask-specific policies for agents, mitigating the potential for homogeneous behaviors effectively. LSVQ exhibits strong portability and allows seamless integration into diverse algorithms, as the subtask learner can be updated concurrently with the MARL framework while requiring only minimal adjustments to the hyperparameters in the joint loss function. Experiments on StarCraft II micromanagement scenarios (SMAC) (Samvelyan et al., 2019) and Google Research Football (GRF) (Kurach et al., 2020) as well as subsequent ablation studies have demonstrated the prominent superiority of our proposed method.

## 2 PRELIMINARIES

### 2.1 PROBLEM FORMULATION

A fully cooperative multi-agent system (MAS) is typically represented by a decentralized partially observable Markov decision process (Dec-POMDP) Oliehoek & Amato (2016), which is composed of a tuple $G = \langle \mathcal{S}, \mathcal{U}, \mathcal{P}, \mathcal{Z}, r, \mathcal{O}, n, \gamma \rangle$. At each time-step, the current global state of the environment is denoted by $s \in \mathcal{S}$, while each agent $a \in \mathcal{A} := \{1, \ldots, n\}$ only receives a unique local observation $z_a \in \mathcal{Z}$ generated by the observation function $\mathcal{O}(s, a) : \mathcal{S} \times \mathcal{A} \to \mathcal{Z}$. Subsequently, every agent $a$ selects an action $u_a \in \mathcal{U}$, and all individual actions are combined to form the joint action $\boldsymbol{u} = [u_1, \ldots, u_n] \in \boldsymbol{\mathcal{U}} \equiv \mathcal{U}^n$. The interaction between the joint action $\boldsymbol{u}$ and the current state $s$ leads to a change in the environment to state $s'$ as dictated by the state transition function $\mathcal{P}(s'|s, \boldsymbol{u}) : \mathcal{S} \times \mathcal{U} \times \mathcal{S} \to [0, 1]$. All agents in the Dec-POMDP share the same global reward function $r(s, \boldsymbol{u}) : \mathcal{S} \times \mathcal{U} \to \mathbb{R}$, and $\gamma \in [0, 1)$ represents the discount factor.

**Definition 1 (Subtasks)** *The cooperative multi-agent task $G = \langle \mathcal{S}, \mathcal{U}, \mathcal{P}, \mathcal{Z}, r, \mathcal{O}, n, \gamma \rangle$ invokes a set of $K$ subtasks $\Phi := \{1, ..., K\}$, where $K$ is set manually. Each subtask $k$ holds a tuple $\langle e_k, \mathcal{A}_k, \pi_k \rangle$, where $e_k \in \mathbb{R}^m$ is the embedding vector of subtask $k$, and $\mathcal{A}_k$ is a set of agents assigned with subtask $k$, satisfying $\cup_{k=1}^{K} \mathcal{A}_k = \mathcal{A}$ and $\mathcal{A}_i \cap \mathcal{A}_j = \emptyset$ for $\forall 1 \leq i < j \leq K$. Each $a_i \in \mathcal{A}_k$ shares the policy network $\pi_k$.*

On the basis of introducing subtasks into Dec-POMDP, each agent $a$ selects a subtask $\phi_k$ based on its own action-observation history $\tau_a \in T \equiv (\mathcal{Z} \times \mathcal{U})$, thus the policy of each agent $a$ can be written as $\pi_k(u_a|\tau_a) : T \times \mathcal{U} \to [0, 1]$. The joint action-value function can be computed by the following formula: $Q^{\boldsymbol{\pi}}(s_t, \boldsymbol{u}_t) = \mathbb{E}_{s_{t+1:\infty}, \boldsymbol{u}_{t+1:\infty}} [R_t|s_t, \boldsymbol{u}_t]$, where $\boldsymbol{\pi}$ is the joint policy of all agents. The goal is to maximize the discounted return $R^t = \sum_{l=0}^{\infty} \gamma^l r_{t+l}$.

### 2.2 VALUE DECOMPOSITION METHOD

Credit assignment is a key problem in cooperative MARL problems. When agents share a joint value function, it is challenging for an individual agent to discern its impact on the collective performance. Insufficient feedback raises the probability of learning failure.

Value decomposition methods assume that each agent has a specific value function for decision-making. The integration of these individual functions creates the joint value function. To guarantee that the optimal action for each agent aligns with the global optimal joint action, all value decomposition methods satisfy the *Individual Global Max* (IGM) Rashid et al. (2018) conditions, which are described below:

$$\arg \max_{\boldsymbol{u}} Q_{tot}(\boldsymbol{\tau}, \boldsymbol{u}) = \begin{pmatrix} \arg \max_{u_1} Q_1(\tau_1, u_1) \\ \vdots \\ \arg \max_{u_n} Q_n(\tau_n, u_n) \end{pmatrix},$$

where $Q_{tot} = f(Q_1, ..., Q_n)$, $Q_1, ..., Q_n$ denote the individual Q-values, and $f$ is the mixing function.

QMIX, the most well-known value decomposition algorithm, is highly regarded for its effectiveness across a range of scenarios. To fulfill the IGM conditions, QMIX confines the parameters of the value mixing network to non-negative values. This is achieved by satisfying the inequality as listed below:

$$\frac{\partial Q_{tot}(\boldsymbol{\tau}, \boldsymbol{u})}{\partial Q_a(\tau_a, u_a)} \geq 0, \quad \forall a \in \{1, \ldots, n\}.$$

### 2.3 VECTOR QUANTISED-VARIATIONAL AUTOENCODER

Vector quantization (VQ) (Gray, 1984) is a classical quantization technique from signal processing and has been widely used for lossy data compression, pattern recognition, density estimation, and clustering. VQ works by encoding values from a high-dimensional vector space into a finite set of values (known as *codebook*) from a lower-dimensional discrete subspace.

Traditional works about generative models, including generative adversarial networks (GAN) (Goodfellow et al., 2014) and variational autoencoder (VAE) (Kingma & Welling, 2013), focus on learning representations with continuous features. However, discrete representations are more suitable for certain modalities (e.g., symbolized language is inherently discrete). Van Den Oord et al. (2017) propose VQ-VAE, in which VQ is adopted to quantize a feature representation layer and VAE has been developed for modeling distributions over discrete latents. Specifically, given an input $x$ which is passed through an encoder producing output $z_e(x)$, the discrete latent variable $z$ is calculated by a nearest neighbor look-up in the predefined codebook $\boldsymbol{E} \in \mathbb{R}^{K \times D}$ which contains $K$ vectors $e_i \in \mathbb{R}^D, i = 1, 2, ..., K$, as shown in Equations 1 and 2.

$$q(z = k|x) = \begin{cases} 1 & \text{for} \quad k = \arg \min_i \|z_e(x) - e_i\|_2, \\ 0 & \text{otherwise} \end{cases}, \tag{1}$$

$$z_q(x) = e_k, \quad \text{where} \quad k = \arg \min_i \|z_e(x) - e_i\|_2, \tag{2}$$

where $q(z|x)$ is a categorical distribution and $z_q(x)$ is the input for the decoder. VQ-VAE gets comparable performance with its continuous counterparts, while it is easier to train and free from the "posterior collapse" issue.

## 3 METHOD

A group of people with assigned responsibilities always establish efficient collaboration in real-life tasks. In MARL, agents with specific subtasks also demonstrate better coordination. In this section, we introduce LSVQ, a novel cooperative MARL framework founded on the concept of subtasks. Instead of using prior knowledge, LSVQ aims to learn a vector representation for each subtask and automatically select a subtask for each agent, which renders the algorithm applicable to diverse scenarios. We first discuss how to discover a set of representative subtasks from the environment. Next, we present how the learned subtasks facilitate the decision and coordination of the agents. The overall optimization objective is presented at the end.

### 3.1 SUBTASK LEARNER

**Assumption 1** *It is sufficient to determine each agent's assigned subtask $z_a$ based on the current global state $s$.*

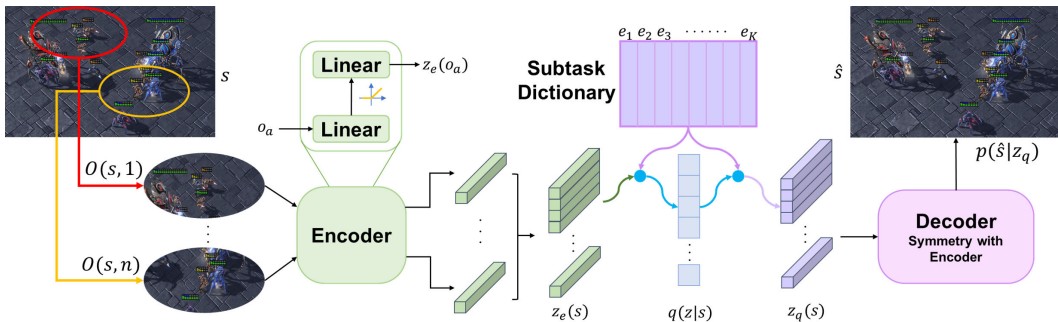

Figure 1: The overall framework of subtask learner.

Following assumption 1, we need to learn a posterior categorical distribution $q(\boldsymbol{z}|s)$ for selecting subtasks. However, only the local observation $o_a$ is available for each agent $a$ under the CTDE paradigm. As $o_a = \mathcal{O}(s, a)$ can be viewed as a projection of the global state $s$, we adopt the following approximation in practice:

$$\arg\max_{\boldsymbol{z}} q(\boldsymbol{z}|s) \approx \begin{pmatrix} \arg\max_{z_1} q(z_1|o_1) \\ \vdots \\ \arg\max_{z_n} q(z_n|o_n) \end{pmatrix} \tag{3}$$

Making decisions solely based on local observation is highly limiting for agents, and a well-learned subtask representation is expected to be informative enough to help infer the global state and enhance coordination. Such an objective can be formulated as below:

$$\max \mathbb{E}_{\boldsymbol{z} \sim q(\boldsymbol{z}|s)} \left[ \log p(s|\boldsymbol{z}) \right] \tag{4}$$

It is intuitive to associate the optimization target in Equation 4 with VAEs. Nevertheless, the subtasks of the agents, which function as the latent variables in VAE, are discrete in nature rather than continuous. Inspired by VQ-VAE (Van Den Oord et al., 2017), we propose a novel subtask learner as Figure 1 shows. This model is sequentially composed of an encoder, a subtask dictionary, and a decoder. The observation $o_a$ of each agent $a$ is fed into the model and interpreted as a perspective of the global state $s$, which is similar to the "channel" in convolutional neural networks. The encoder $z_e(\,\cdot\,;\theta_e)$, consisting of two fully-connected layers and a ReLU activation unit, encodes the observation $o_a$ into a vector $z_e(o_a) \in \mathbb{R}^D$ of equivalent length to the subtask representation.

The subtask dictionary maps the $K$ subtasks to representations $\boldsymbol{E} = \{e_1, ..., e_K\}$. Each agent $a$ selects its subtask based on a deterministic distribution $q(z|o_a)$ that is computed using embedding $z_e(o_a)$ and subtask representations, in accordance with Equation 1. Meanwhile, the output of the encoder $z_e(o_a)$ is transformed into $z_q(o_a) \sim q(z|o_a)$ by the discretization bottleneck according to Equation 2.

Following the approximation in Equation 3, we aggregate all the $z_q(o_a)$ together and deduce the latent representation of the global state $z_q(s)$. Next, the model passes $z_q(s)$ through a decoder $f_d(\,\cdot\,;\theta_d)$ that is symmetric in structure to the encoder, in order to reconstruct the original global state as below:

$$\hat{s} = f_d\left(z_q(s)\right) = f_d\left(\text{concatenate}\left(z_q(o_1), ..., z_q(o_n)\right)\right) \tag{5}$$

Typically, a VAE is updated by maximizing the evidence lower bound (ELBO) (Kingma & Welling, 2013), which consists of a reconstruction loss term and a Kullback-Leibler (KL) divergence term. Similar but not the same, the overall loss function of our subtask learner is given by Equation 6, in which sg stands for stopping the propagation of gradients. The first term is the state reconstruction loss which optimizes the decoder and encoder. We suppose a uniform distribution $p(z) = \frac{1}{K}$ as the prior for all subtasks, and the KL divergence term in ELBO can be ignored as it is a constant:

$$D_{KL}\left(q(z|o_a)\|p(z)\right) = 1 \cdot \log\left(\frac{1}{\frac{1}{K}}\right) + (K-1) \cdot 0 \cdot \log\left(\frac{0}{\frac{1}{K}}\right) = \log K,$$

where $D_{KL}$ refers to KL divergence. However, the representations $\boldsymbol{E}$ in the subtask dictionary receive no gradients from the first term. Therefore, we employ the vector quantization objective in dictionary learning in the second term, which moves the chosen representation $e_k$ towards the embedding $z_e(x)$ by minimizing the $l_2$ error. The last term serves as a regularizer for the encoder. It encourages the output of the encoder to stay close to the chosen representation to prevent oscillations in the agents' subtask selection.

$$\mathcal{L}_{sub} = \|\hat{s} - s\|_2^2 + \sum_{a=1}^{n} \|\text{sg}\left[z_e(o_a)\right] - z_q(o_a)\|_2^2 + \beta \sum_{a=1}^{n} \|z_e(o_a) - \text{sg}\left[z_q(o_a)\right]\|_2^2 \qquad (6)$$

## 3.2 LSVQ ARCHITECTURE

Because the subtask representations are able to capture the global state information, it is imperative to integrate them into the decision-making process of agents. Figure 2 depicts the overview of classic MARL methods with our proposed subtask learner. First, the subtask learner assigns a subtask $q_a$ to agent $a$ using its local observation. Next, both the local information and the assigned subtask are fed into the Q network in Figure 2(a) or the policy network in Figure 2(b) to produce the action at the current time step. Instead of simply concatenating the local information with the assigned subtask, we refer to the implementation in LDSA (Yang et al., 2022) and employ hypernetworks (Ha et al., 2016) to build a subtask-specific policy for each agent. Consider the model in Figure 2(a) as an example, each agent $a$ uses a shared trajectory encoder composed of a multi-layer perceptron (MLP) and a gated recurrent unit (GRU) (Chung et al., 2014) to process the local observation and history preliminarily. This encoder is followed by another MLP, of which the parameters are generated by a hypernetwork that takes the subtask representation $e_{z_a}$ as input. At every time step, agent $a$ can adopt one of the $K$ distinct subtask-specific Q networks for decision-making according to its selected subtask, leading to a more flexible and versatile policy.

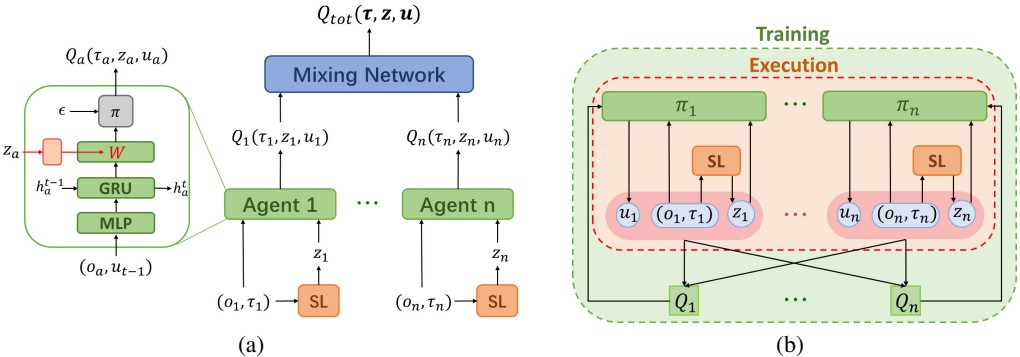

Figure 2: Illustration of MARL methods with subtasks. SL refers to the subtask learner. **Left:** The value decomposition method with subtask learner. **Right:** The multi-agent actor-critic method with subtask learner.

LSVQ makes modifications to the decision-making process of agents while leaving other parts of the model unchanged, such as the mixing network in QMIX (Rashid et al., 2018) and the critic networks in MADDPG (Lowe et al., 2017). Therefore, LSVQ can be readily deployed into various MARL algorithms for cooperative tasks. For value decomposition methods with LSVQ, the model is trained by minimizing the following loss:

$$\mathcal{L}_{RL} = \left(y_{tot} - Q_{tot}(\boldsymbol{\tau}, \boldsymbol{z}, \boldsymbol{u})\right)^2, \qquad (7)$$

where $y_{tot} = r + \gamma \max_{\boldsymbol{u}'} \hat{Q}_{tot}(\boldsymbol{\tau}', \boldsymbol{z}', \boldsymbol{u}')$. In the multi-agent policy gradient methods with LSVQ, the critic network $Q_a(s, z_a, u_1, ..., u_n)$ is updated by a loss similar to Equation 7, and the policy gradient is calculated as:

$$\nabla J(\pi_a) = \mathbb{E}_{\mathcal{D}}\left[\nabla \pi_a(\tau_a, z_a)\nabla_{u_a}Q_a(s, z_a, u_1, ..., u_n)|u_a = \pi_a(\tau_a, z_a)\right]. \qquad (8)$$

The overall optimization objective is simply the sum of $\mathcal{L}_{sub}$ and $\mathcal{L}_{RL}$, as shown in Equation 9. In our implementation, the gradients of reinforcement learning are detached from the subtask learner to maintain the independence of distinct modules during training, hence the model can be trained end-to-end without the need to manually adjust the weights of two loss terms for an equilibrium. It is worth noting that $\mathcal{L}_{sub}$ is also insensitive to coefficient $\beta$ in the range from 0.1 to 2, and the only extra hyperparameter that needs to tune in LSVQ compared to the original method is the number of subtasks $K$. These merits enable the facile migration of LSVQ onto a broad spectrum of MARL algorithms.

$$\mathcal{L} = \mathcal{L}_{RL} + \mathcal{L}_{sub} \tag{9}$$

## 4 EXPERIMENTS

We evaluate the effectiveness of LSVQ on two popular testbeds for multi-agent cooperative tasks. In our experiments, we primarily employ LSVQ-QMIX, a variant that deploys LSVQ into the QMIX framework, owing to the prevalent conjunction of previous subtask-based MARL methods with value decomposition methods (Wang et al., 2020b;c; Yang et al., 2022; Zeng et al., 2023). A number of recognized state-of-the-art (SOTA) methods have been selected as baselines, including QMIX (Rashid et al., 2018), MAVEN (Mahajan et al., 2019), and three subtask-based algorithms (ROMA (Wang et al., 2020b), RODE (Wang et al., 2020c), LDSA (Yang et al., 2022)). All of the above methods possess an identical mixing network structure and differ from the architecture of their respective agent networks. Therefore, the performance of LSVQ can be assessed intuitively and impartially. The algorithms in our experiments are implemented based on the PyMARL (Samvelyan et al., 2019), without any modification made to the original environment settings or hyperparameters.

### 4.1 PERFORMANCE ON MULTI-AGENT ENVIRONMENTS

StarCraft Multi-Agent Challenge (SMAC) (Samvelyan et al., 2019) offers a diversity of micro-war scenarios in StarCraft II, in which each unit has limited local observation and a large action space. Notably, the version of StarCraft II used in our experiments is 4.6.2 instead of the simpler 4.10, and results from different versions are not always comparable. We conducted experiments on a group of representative scenarios of various difficulty levels, and Figure 3 shows the results on six *Hard* and *Super Hard* scenarios, in which the $25 - 75\%$ percentiles are shaded. In *3s_vs_5z* and *corridor*, RODE performs slightly better than LSVQ-QMIX, but it may fail in certain scenarios. Meanwhile, LSVQ-QMIX exhibits the best overall performance and outperforms all baselines in most scenarios. LSVQ makes significant progress on the basis of QMIX, as LSVQ-QMIX not only elevates the performance but also effectively tackles challenges that QMIX is unable to solve. This demonstrates its great generality and superior performance in multi-agent coordination tasks, and we believe the reason is that LSVQ promotes cooperative behaviors by providing agents with informative subtasks.

Google Research Football (GRF) (Kurach et al., 2020) offers a user-friendly platform for training and evaluating agents in a football game, which has been well received by the reinforcement learning community. Agents are required to refine their individual skills as well as learn to cooperate with teammates. The performance and analysis of LSVQ-QMIX on GRF scenarios can be found in Appendix D.1.

### 4.2 PRE-TRAINED SUBTASK LEARNER

Here we provide another paradigm for training LSVQ-QMIX, which can be extended to other variants. We first use a policy $\mu$ to collect data from the environment and train the subtask learner in light of Equation 6. Then the parameters in the subtask learner are fixed, and we update the joint policy $\pi$ following Equation 7. We denote this version as LSVQ_PT and test it on three SMAC scenarios. From Figure 4 we find that LSVQ_PT exhibits a similar learning process and lower variance in comparison to LSVQ-QMIX, which proves the feasibility of the pre-trained subtask learner.

However, due to the discrepancy between policy $\pi$ and $\mu$, the subtask learner receives data with distinct distributions during training and execution. This issue is frequently encountered in offline reinforcement learning and may have a detrimental impact on algorithm performance. Some studies

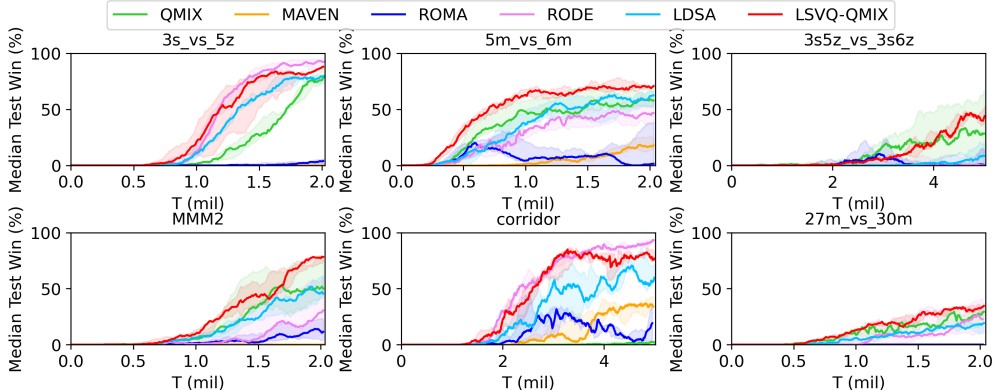

Figure 3: Comparison of LSVQ-QMIX against baselines on SMAC scenarios.

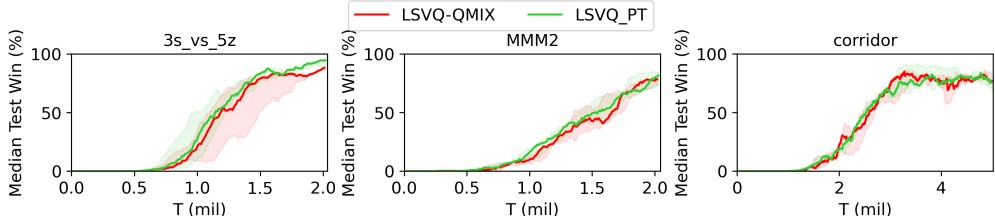

Figure 4: Comparison of different training methodologies.

suggest that training the subtask learner and the MARL framework alternately can alleviate the problem and improve the results (Hafner et al., 2019).

### 4.3 VISUALIZATION OF SUBTASKS

To investigate how the subtask learner works and affects the behaviors of agents, we first employ t-SNE (Van der Maaten & Hinton, 2008) for dimensionality reduction to visualize the raw observations and the associated embeddings within the subtask learner. As shown in Figure 5(a) and 5(b), the observations associated with the same subtask tend to exhibit proximity within the 2D t-SNE embedding space, and the clarity of clusters is enhanced after encoding the observations into latent subtask space. This evidence indicates that the subtask learner can derive informative representations by discovering and accentuating potential correlations from individual observations of agents.

Next, we select an episode to conduct a case study. Figure 5(c) depicts the subtasks assigned to agents within the episode, and the related game screens are displayed in Figure 5(d). At the beginning, agents are arranged in a formation with two wings spread out, and the Medivac is protected at the rear. Later, the injured agents retreat to the back of the formation while the healthy ones move forward. After gaining a numerical advantage, the allies focus fire to kill the rest enemies one by one. We notice that the attacking agents tend to select the subtask in orange while the dead ones opt for the subtask in green. Thanks to vector quantization, LSVQ enables agents to select various subtasks in different stages and situations as well as preventing too frequent changes of chosen subtasks that could lead to unstable training, whereas other subtask-based methods require additional regularization terms and hyperparameters to achieve the same.

### 4.4 ABLATION

We investigate three issues below: (a) Is the superior performance of LSVQ from the subtasks or the additional parameters? (b) Can LSVQ be transferred to other MARL frameworks? (c) How does the number of subtasks $K$ influence the performance of LSVQ?

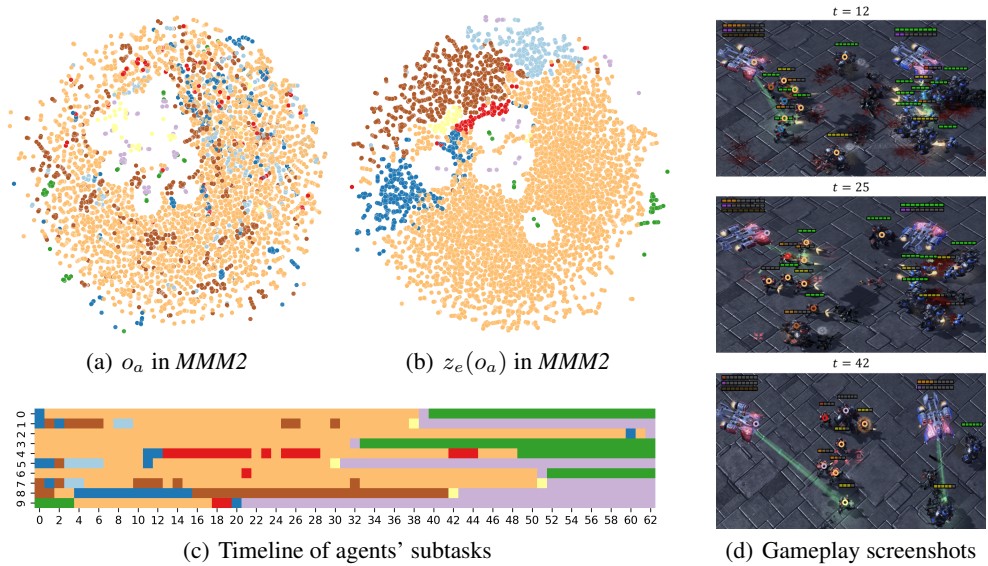

(a) $o_a$ in *MMM2*    (b) $z_e(o_a)$ in *MMM2*

(c) Timeline of agents' subtasks    (d) Gameplay screenshots

Figure 5: Visualization of subtasks in *MMM2* scenario. (a)(b) The 2D t-SNE embedding of the observations $o_a$ and the subtask encoder's outputs $z_e(o_a)$ respectively. (c) The changes of subtasks assigned to each agent in an episode. (d) The game screenshots of the episode, in which each agent is marked with a colored circle representing its assigned subtask. The alignment of colors and subtasks is consistent across figures.

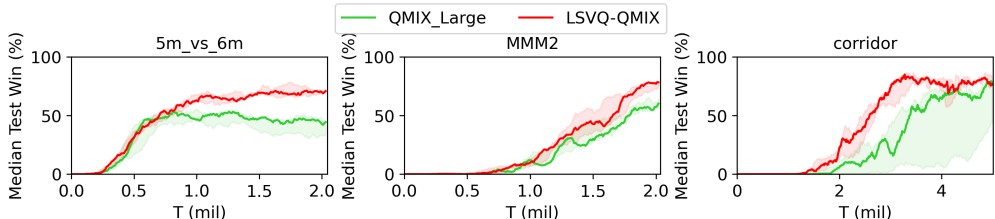

Figure 6: Ablation study on the number of parameters.

First, we implement QMIX-Large, within which the scale of the Q network is enlarged to achieve a comparable parameter count with that of LSVQ-QMIX. Results in Figure 6 have indicated that simply adding parameters does not make a significant improvement to QMIX. On the contrary, the large network imposes a great burden on the training process, leading to a decline in performance in some cases.

Second, we integrate LSVQ with MADDPG (Lowe et al., 2017) and test it in three cooperative scenarios with the same continuous action space and different numbers of agents: *Continuous_pred_prey_3a*, *6a* and *9a* (Peng et al., 2021), in which we control a team of predators to catch the preys with built-in heuristic policies. The details of the scenarios can be found in Appendix B.4. The hyperparameter configurations of LSVQ-MADDPG remain exactly the same as MADDPG, without any adjustments. According to the learning curves in Figure 7, LSVQ-MADDPG can successfully solve all the tasks and achieve results equivalent to or even superior to those of MADDPG. This strongly demonstrates the capability of LSVQ to transfer to other algorithms and problems.

Last, we compare the performances of LSVQ-QMIX with different numbers of subtasks while keeping other settings unchanged. We represent the variants as LSVQ($K$), in which $K$ refers to the number of subtasks. Figure 8 displays the outcomes in two distinct scenarios. A large value of $K$ does not necessarily result in improved performance, as an excessive number of subtasks can increase the scale of the problem and impede coordination. The key is to choose an appropriate $K$ based on the overall task to decompose and reach a balance between policy diversity and model complexity.

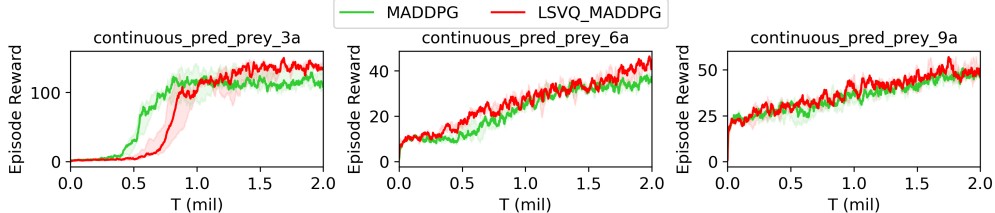

Figure 7: The performance of LSVQ-MADDPG with continuous action space.

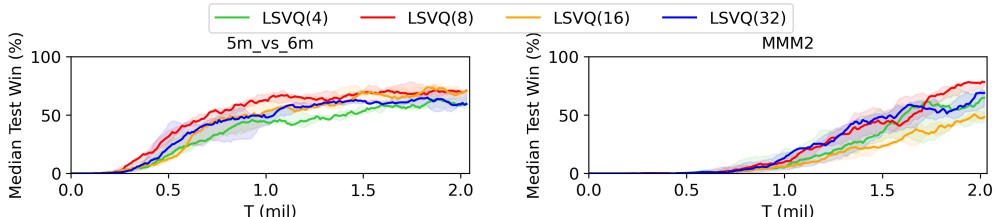

Figure 8: Ablation study on the influence of the number of subtasks.

## 5  CONCLUSION

To overcome the limitations of previous subtask-based MARL methods, we propose a novel learning framework called LSVQ. Inspired by VQ-VAE, LSVQ designs a subtask learner that attempts to reconstruct the global state based on discrete latent variables. Therefore, the latent variables possess the ability to reflect a broader context and serve as informative subtask representations. By incorporating LSVQ with classic MARL methods, the coordination of agents is significantly improved with the aid of assigned subtasks. We hope that LSVQ could provide a scalable and transferable approach for addressing general cooperative problems in multi-agent systems.

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

## A    RELATED WORKS

### A.1    VALUE DECOMPOSITION METHODS

Value decomposition is an effective technique for solving the credit assignment problem and has gained popularity in recent years. It factorizes the global Q-value into individual Q-values following the individual-global-max (IGM) assumption (Rashid et al., 2018). VDN (Sunehag et al., 2017) and QMIX (Rashid et al., 2018) factorize the global Q-value by additivity and monotonicity, respectively. Qatten (Yang et al., 2020) replaces the MLP in the mixing network of QMIX with the attention mechanism. QTRAN (Son et al., 2019) transforms the original global Q-value function into another one that is easier to decompose and equivalent to the original task. Weighted QMIX (Rashid et al., 2020) adjusts the weights of different joint actions when updating to overcome the limitation imposed by the monotonic constraint in QMIX. QPLEX (Wang et al., 2020a) proposes a duplex dueling network architecture to implement the complete IGM function class.

### A.2    MULTI-AGENT POLICY GRADIENT

Multi-agent policy gradient algorithms enjoy stable theoretical convergence properties, and hold the promise to extend MARL to continuous control problems. COMA (Foerster et al., 2018) proposes the paradigm of centralized critic and decentralized actor (CCDA). MADDPG (Lowe et al., 2017) and MAPPO (Yu et al., 2022) migrate the original algorithms to multi-agent scenarios. MAAC (Iqbal & Sha, 2019) extends CCDA by introducing the attention mechanism. LICA (Zhou et al., 2020) proposes a global critic that takes the global state and the joint action as input to calculate a global value estimation like value decomposition methods do.

### A.3    SUBTASK-BASED METHODS

By decomposing the overall task, agents can focus on restricted subtasks that are easier to solve. Besides, a proper subtask assignment can facilitate the cooperative behaviors of agents. However, it is challenging to come up with a set of subtasks that can effectively decompose the whole multi-agent task. The most straightforward way is to predefine the subtasks by leveraging the prior domain knowledge(Becht et al., 1999; Stone & Veloso, 1999; Spanoudakis & Moraitis, 2010; Lhaksmana et al., 2018). Recent methods attempt to automatically decompose the task without prior knowledge. ROMA (Wang et al., 2020b) introduces the concept of roles for each agent based on its local observation and conditions agents' policies on their roles. RODE (Wang et al., 2020c) explicitly defines the subtasks based on joint action space decomposition during pretraining, where each subtask is mapped on a subset of actions. RODE is considered to be a SOTA method due to its superior performance, but it would fail in certain scenarios as it restricts the available actions of the agent after it is assigned a subtask. LDSA (Yang et al., 2022) builds subtask representations from one-hot vectors, then it generates subtask-specific Q networks for decision-making. The subtask in LDSA is used to control the diversity of agents' behavior, but it does not necessarily improve coordination as there is no global information in the subtask representations to guide the agents.

## B    ENVIRONMENT DETAILS

### B.1    SMAC

SMAC is a simulation environment for research in collaborative multi-agent reinforcement learning (MARL) based on Blizzard's StarCraft II RTS game. The goal of each task is to control different types of agents to move or attack to defeat the enemies. The version of StarCraft II is 4.6.2 (B69232) in our experiments, and it should be noted that results from different client versions are not always comparable. The difficulty of the game AI is set to *very hard* (the 7th level). Table 1 presents the details of selected scenarios in our experiments.

### B.2    SMACV2

SMACv2 (Ellis et al., 2023) is proposed to address SMAC's lack of stochasticity. In SMACv2, the team compositions and agent start positions are generated randomly at the beginning of each

Table 1: Information of selected challenges.

| Challenge | Ally Units | Enemy Units | Type | Level of Difficulty |
|---|---|---|---|---|
| 3s_vs_5z | 3 Stalkers | 5 Zealots | Homogeneous Asymmetric | Hard |
| 5m_vs_6m | 5 Marines | 6 Marines | Homogeneous Asymmetric | Hard |
| 3s5z_vs_3s6z | 3 Stalkers 5 Zealots | 3 Stalkers 6 Zealots | Heterogeneous Asymmetric | Super Hard |
| MMM2 | 1 Medivac 2 Marauders 7 Marines | 1 Medivac 3 Marauders 8 Marines | Heterogeneous Asymmetric | Super Hard |
| corridor | 6 Zealots | 24 Zerglings | Homogeneous Asymmetric | Super Hard |
| 27m_vs_30m | 27 Marines | 30 Marines | Homogeneous Asymmetric | Super Hard |

episode. The sight range of the agents is narrowed, and the attack ranges of different unit types are no longer the same. These modifications make SMACv2 extremely challenging. The version of the StarCraft II engine is also 4.6.2 (B69232) in our experiments.

### B.3 GOOGLE RESEARCH FOOTBALL

We choose three official scenarios from Football Academy. *Academy_3_vs_1_with_keeper* and *Academy_pass_and_shoot_with_keeper* are relatively easy scenarios, and *Academy_Corner* is a hard one. Agents are rewarded when they score a goal or kick the ball to a position close to the goal. Observations of the agent include the relative positions of all other entities. We restrict the football to the opponent's half of the field and stop the episode once the ball enters our half to speed up training.

### B.4 COOPERATIVE PREDATOR-PREY

Based on the mixed *simple tag* scenario in multi-agent particle environments (MPE), we modify it into a cooperative task named *continuous_pred_prey*, following the implementation of Peng et al. (2021). The modified scenario is depicted in Figure 9. The prey is controlled by a heuristic policy which moves the prey to the sampled position with the largest distance to the closest predator. The predators gain a team reward of $+10$ when one of them collides with a prey. To introduce partial observability to the environment, the agents are constrained to access information within a view radius. We extend the task to three variants with different numbers of agents. For example, *continuous_pred_prey_9a* means there are 9 predators and $9/3 = 3$ preys in the environment.

## C IMPLEMENTATION DETAILS

### C.1 SETTINGS OF HYPERPARAMETERS

We list the hyperparameters of the variants of LSVQ in Table 2. The hyperparameters of the other baselines remain the same as their official implementations.

### C.2 ALGORITHMIC DESCRIPTION

We mainly test the performance of LSVQ-QMIX in this paper, and the pseudocode is shown in Algorithm 1. The code for LSVQ can be found in the supplementary material.

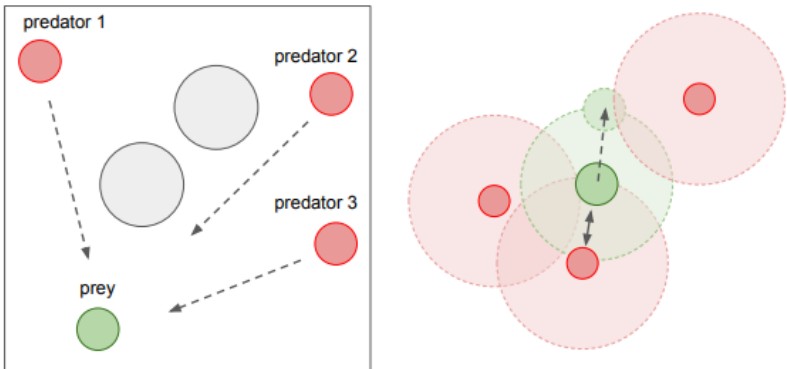

Figure 9: The illustration of cooperative predator-prey scenario. **Left**: The top-down view of the environment, with predators (red), prey (green), and obstacles (grey). **Right**: The observation radii of the agents, and the prey's heuristic of escape.

Table 2: Information of selected challenges.

| Algorithm | Description | Value |
|---|---|---|
| QMIX | Type of optimizer | RMSProp |
| | Exploration mode | $\epsilon$-greedy |
| | Learning rate | 0.0005 |
| | Target network update interval | 200 |
| | Batch size | 32 |
| | Replay buffer size | 5000 |
| | Discount factor $\gamma$ | 0.99 |
| | Probability of random action ($\epsilon$) | $1.0\tilde{0}.05$ |
| | $\epsilon$ anneal time | 50000 |
| MADDPG | Type of optimizer | Adam |
| | Exploration mode | Gaussian |
| | Learning rate | 0.01 |
| | Target soft update $\tau$ | 0.001 |
| | Batch size | 32 |
| | Replay buffer size | 5000 |
| | Discount factor $\gamma$ | 0.85 |
| | Standard deviation of action noise | 0.1 |
| LSVQ | Number of subtasks | 8 |
| | Dimension of subtask representations | 32 |
| | Coefficient of regularizer $\beta$ | 0.4 |

## D ADDITIONAL EXPERIMENTS

Due to the space limitations, we display some additional experiment results here.

### D.1 EXPERIMENTS ON GRF

Figure 10 presents the results on three GRF scenarios. Although the performance of LSVQ-QMIX is inferior to LDSA in *academy_3_vs_1_with_keeper*, it still reaches promising results and maintains its effectiveness while other methods may fail in particular scenarios.

### D.2 EXPERIMENTS ON SMACV2

We test LSVQ on SMACv2 and select QMIX and LDSA as baselines. QMIX is the baseline that performs best in the original paper of SMACv2, and LDSA is an outstanding subtask-based method with a decision-making process similar to LSVQ but lacking global information. Both LSVQ and LDSA employ the mixing network structure of QMIX. The results are displayed in Figure 11. LSVQ

---

**Algorithm 1:** LSVQ-QMIX

---

Initialize replay buffer $\mathbf{D}$
Initialize the subtask encoder $z_e$, decoder $f_d$, and subtask representations $\mathbf{E}$. Initialize the
  QMIX network with parameters $\theta$, initialize target parameters $\theta^- = \theta$
**while** *training* **do**
    **for** $episode \leftarrow 1$ **to** $M$ **do**
        Start with initial state $\mathbf{s}^0$ and each agent's observation $\mathbf{o}_a^0 = \mathcal{O}\left(\mathbf{s}^0, a\right)$
        Initialize an empty episode recorder $\mathbf{R}$ for $t \leftarrow 0$ **to** $T$ **do**
            For every agent $a$, with probability $\epsilon$ select action $u_a^t$ randomly
            Otherwise, calculate $z_q(o_a^t) = e_k \in \mathbf{E}$ and assign $a$ with subtask $z_a^t = k$ based on
              Equation 1 and 2, select $u_a^t = \arg\max_{u_a^t} Q_a\left(\tau_a^t, z_a^t, u_a^t\right)$
            Take joint action $\boldsymbol{u}^t$, and retrieve next state $\mathbf{s}^{t+1}$, next observations $\boldsymbol{o}^{t+1}$ and reward
              $r^t$
            Store transition $\left(\mathbf{s}^t, \boldsymbol{o}^t, \boldsymbol{u}^t, r^t, \mathbf{s}^{t+1}, \boldsymbol{o}^{t+1}\right)$ in $\mathbf{R}$
        **end**
        Store episode data $\mathbf{R}$ in $\mathbf{D}$
    **end**
    Sample a random mini-batch data $\mathbf{B}$ with batch size $N$ from $\mathbf{D}$
    **for** $t \leftarrow 0$ **to** $T - 1$ **do**
        Extract transition $\left(\mathbf{s}^t, \boldsymbol{o}^t, \boldsymbol{u}^t, r^t, \mathbf{s}^{t+1}, \boldsymbol{o}^{t+1}\right)$ from $\mathbf{B}$
        For every agent $a$, calculate $z_q(o_a^t)$ and $Q_a\left(\tau_a^t, z_a^t, u_a^t | \theta\right)$
        Reconstruct $\hat{s}$ in light of Equation 5
        Calculate $Q_{tot}\left(\boldsymbol{\tau}^t, \boldsymbol{z}^t, \boldsymbol{u}^t | \theta\right)$
        With target network, calculate $Q_a\left(\tau_a^{t+1}, z_a^{t+1}, u_a^{t+1} | \theta^-\right) = \max Q_a\left(\tau_a^{t+1}, \cdot | \theta^-\right)$
        Calculate $Q_{tot}\left(\boldsymbol{\tau}^{t+1}, \boldsymbol{z}^{t+1}, \boldsymbol{u}^{t+1} | \theta^-\right)$
    **end**
    Update the subtask learner by minimizing the loss in Equation 6
    Update $\theta$ by minimizing the loss in Equation 7
    Update target network parameters $\theta^- = \theta$ periodically
**end**

---

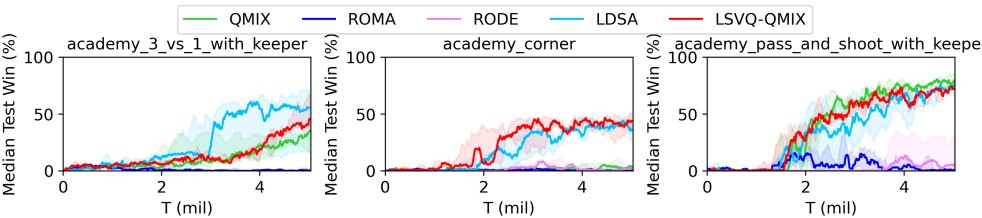

Figure 10: Comparison of LSVQ-QMIX against baselines on GRF scenarios.

achieves the optimal overall outcomes and is superior to LDSA. This indicates that the global information incorporated in the subtask representations is very helpful to the agents' coordination under conditions with severely restricted observations. It also demonstrates the universal effectiveness of LSVQ.

### D.3 VISUALIZATION OF SUBTASK LEARNER

Besides *MMM2*, we also investigate how the subtask learner works in *corridor* to assess the effectiveness of LSVQ in various scenarios. The 2D t-SNE embeddings of the observations are shown in Figure 12(a), and we find that the observations with proximity are classified into the same subtask by LSVQ. According to Figure 12(b), the correlation between observations from the same subtask is further accentuated after being processed by the subtask encoder. This suggests that LSVQ learns reasonable subtask representations from a global perspective.

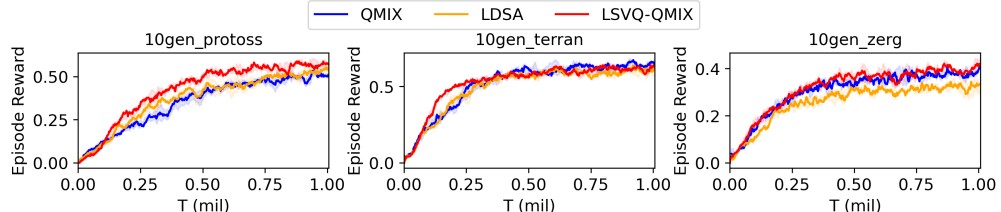

Figure 11: Results on SMACv2 scenarios.

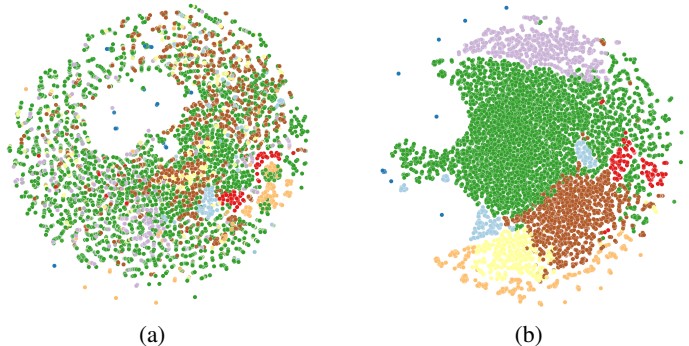

(a)          (b)

Figure 12: Visualization of *corridor* scenario.

