# OpenReview forum: "Constructing Informative Subtask Representations for Multi-Agent Coordination"
_ICLR.cc/2024/Conference — Submitted to ICLR 2024_

### Official Review · Reviewer_E766 · 2023-10-21

**Soundness:** 2 fair
**Presentation:** 2 fair
**Contribution:** 2 fair
**Rating:** 5
**Confidence:** 4

**Summary:**

Automatic subtask decomposition and assignment have a great potential to improve the learning performance and efficiency for multi-agent reinforcement learning (MARL), which is also quite challenging to solve. This paper makes some effort in this direction, but the algorithm design relies on strong assumptions and lacks novelty compared with previous works. The performance improvements on standard benchmarks are limited. Thus, I suggest a rejection.

**Strengths:**

(a) This paper proposes an algorithm direction for targeting a challenging but fruitful research area: automatic subtask decomposition and assignment in MARL.

(b) The proposed definition of subtasks is very insightful.

**Weaknesses:**

(a) Multiple related works on MARL are included for comparison in Section 1, but the intuition behind the proposed algorithm is still not very clear.

(b) The definition of subtasks is a little restricted, since it requires that each agent is assigned with only one subtask.

(c) Assumption 1 is strong, as the information contained in the current state only may not support effective subtask assignment, given that subtask execution usually requires multiple time steps. The approximation in Eq. (3) further assumes that each agent can determines its own subtask based on its local observation, which is not rational since effective subtask assignment requires global knowledge.

(d) The novelty in algorithm design is a little limited, by introducing a subtask learner in QMIX. The subtask learner design is based on VQ-VAE, and the hypernetwork idea has also been adopted in QMIX.

(e) QMIX itself has some drawbacks as it assumes that the global Q-value grows monotonically with individual ones. Following works, like QDPP and Weighted QMIX, have explicitly pointed out this issue and improved upon QMIX.

**Questions:**

(a)  Could you give more explanation on the statement from Section 1: "Nevertheless, these approaches aim to build differentiated subtask representations but neglect the information that ought to be incorporated within the subtask."?

(b) It would be better to include discussion and comparisons with related works on multi-agent skill (a.k.a., option) discovery.

(c) Could you explain why $q(z|o_a)$ is designed to be deterministic rather than a distribution on all subtasks?

(d) The results reported in the original paper of LDSA are clearly better than the ones shown in this paper, e.g. results on "5m_6m" and "3s5z_3s6z".

---

> ### Author Response · Authors · 2023-11-21
>
> Thanks for your comments. LSVQ is a generic method that can be integrated with various MARL algorithms. As shown in Figure 2, the backbone MARL methods can be QMIX as well as Weighted QMIX, QPLEX, MAPPO, LICA, and so on. LSVQ introduces a subtask learner, which selects a subtask for the agent based on its observation. Then, LSVQ modifies the structure of the policy network, which additionally takes the agent's subtask as input to facilitate coordination. Under the setting of Dec-POMDP, LSVQ has to do subtask assignment with only individual observations available. Therefore, LSVQ maps the observation to latent space, and uses the latent vectors to reconstruct the true global state during training, which can be viewed as maximizing the mutual information $\mathcal{I}(s;\boldsymbol{z})$. In this way, the global information is injected into the latent vectors, and agents can infer the global state by selecting subtask $z_i$ based on its observation.
>
> Q1: Could you give more explanation on the statement from Section 1: "Nevertheless, these approaches aim to build differentiated subtask representations but neglect the information that ought to be incorporated within the subtask?
>
> A1: We assert that a good subtask representation should be informative. This objective can be formalized as maximizing the mutual information $\mathcal{I}(s;\boldsymbol{z})$, which implies that the subtask representation can help to infer the global state. Only with the guide of global information can agents divide up the task and cooperate effectively. In previous works like LDSA, the subtask encoder is updated via complex loss function to make the representations of different subtasks distinguishable, but there is no global information included in to facilitate agents' coordination. This is where LSVQ makes improvements.
>
> Q2: It would be better to include discussion and comparisons with related works on multi-agent skill (a.k.a., option) discovery.
>
> A2: We briefly introduce some related subtask-based studies in Section 1. A more detailed related work section will be included in the future edition.
>
> Q3: Could you explain why $q(z|o_a)$ is designed to be deterministic rather than a distribution on all subtasks?
>
> A3: We calculate $q(z|o_a)$ deterministically because we adopt vector quantization to learn the subtask representations. The representations of $K$ subtasks are saved in a codebook with shape $K\times D$ where $D$ is the dimension of representation. The observation $o_a$ is quantized to a $D$-dimensional vector, and the index of its nearest neighbor vector in the codebook corresponds to the selected subtask. This step of calculating the nearest neighbor makes $q(z|o_a)$ deterministic. Vector quantization offers many benefits. It guarantees distinguishable subtask representations, without the need for an additional loss function to control the discrepancy between them like other subtask-based methods do. It enables agents to select various subtasks in different stages and situations, as well as prevents the agent from frequently changing the selected subtasks in an episode, which could lead to unstable training.
>
> Q4: The results reported in the original paper of LDSA are clearly better than the ones shown in this paper, e.g. results on "5m_6m" and "3s5z_3s6z".
>
> A4: We tested LDSA with the official code released on Openreview. The difference is that in the original paper, the version of the StarCraftII game engine is 4.10. In our experiments, we use version 4.6.2 instead. Experiments in previous studies show that version 4.6.2 is more challenging, with a drop in algorithm performance observed in most scenarios. This is the reason for the discrepancy in the results of LDSA.

---

> > ### Comment · Reviewer_E766 · 2023-11-22
> >
> > Thank you for the detailed response. Still, I think the paper has great potential but needs improvements in the algorithm design. Empirically, the improvements brought by LSVQ are not significant, such as Fig. 7, Fig. 10, and Fig. 11. I would suggest involving more information for the sub-task decomposition and integrating LSVQ with multiple MARL algorithm to show the broad applicability and effectiveness of LSVQ. I would slightly improve my score.

---

### Official Review · Reviewer_J7bq · 2023-10-30

**Soundness:** 2 fair
**Presentation:** 3 good
**Contribution:** 3 good
**Rating:** 5
**Confidence:** 4

**Summary:**

The paper focuses on subtask learning for cooperative multi-agent reinforcement learning (MARL). A variational autoencoder variant is employed to reconstruct the global state from individual observations. The latent encoder vector is discretized via vector quantization before being used by the decoder for reconstruction. The learned latent vectors are used to generate individual policies for all agents belonging to a subtask, using hypernetworks. The approach, called Learn Subtask representations via Vector Quantization (LSVQ), can be integrated into any MARL framework. It is evaluated in a variety of settings in SMAC, particle environments, and Google Research Football and compared with state-of-the-art MARL approaches.

**Strengths:**

The paper proposes an interesting approach to cooperative MARL.

The paper is well-written and easy to understand. I also like the illustrations in Figures 1 and 2 that make the approach better understandable.

**Weaknesses:**

**Novelty**

While the approach seems somewhat novel to me, I am missing a related work section that discusses conceptual differences to other works that encode observations with alternative techniques, e.g., attention, or dynamically group agents for value decomposition such as (in addition to all the works briefly listed in the introduction):

[1] S. Iqbal et al., “Actor-Attention-Critic for Multi-Agent Reinforcement Learning”, ICML 2019

[2] T. Phan et al., “VAST: Value Function Factorization with Variable Agent Sub-Teams”, NeurIPS 2021

[3] M. Wen et al., "Multi-Agent Reinforcement Learning is A Sequence Modeling Problem", NeurIPS 2022

**Soundness**

In contrast to stated in the paper, general Dec-POMDPs have **stochastic** observations [4,5] and are therefore sampled from a distribution instead of being projected via a deterministic function. The benchmark domains used in the paper are all special cases with deterministic observations (and even initial states) thus do not reflect the general characteristics of a general Dec-POMDP [5].

[4] F. Oliehoek et al., "A Concise Introduction to Decentralized POMDPs", 2016

[5] X. Lyu et al., "A Deeper Understanding of State-Based Critics in Multi-Agent Reinforcement Learning", AAAI 2022

According to [4], the global state cannot be reconstructed from observations alone. However, in Dec-MDPs (a special case of Dec-POMDP), the reconstruction is possible. Thus, I am not sure about the general validity of the proposed approach.

**Significance and Clarity**

The x-axes vary for different scenarios in Figures 3, 4, and 6 (either 2 million steps or 5 million), which is somewhat confusing. I wonder how the plots would look like, if the plots with only 2 million steps were run until 5 million steps as there could be a chance that the baselines outperform LSVQ in the long run.

In the maps `5m_vs_6m`, `MMM2`, and `27m_vs_30m`, QMIX performs worse than reported in the original introduction of the benchmark [7]. Furthermore, since the original SMAC authors consider SMAC to be outdated themselves [8], I suggest to evaluate on the newer SMACv2, which exhibits more stochasticity and better aligns with the general Dec-POMDP setting.

[7] M. Samvelyan et al, "The StarCraft Multi-Agent Challenge", AAMAS 2019

[8] B. Ellis et al., "SMACv2: An Improved Benchmark for Cooperative Multi-Agent Reinforcement Learning", 2022

**Minor**

- Some plots only show red and green lines. This is bad for colorblind readers who cannot distinguish between those lines. I suggest to change one of these colors to blue.

**Questions:**

None

---

> ### Author Response · Authors · 2023-11-21
>
> Q1: The reviewer is missing a related work section that discusses conceptual differences to other works that encode observations with alternative techniques, e.g., attention, or dynamically group agents for value decomposition.
>
> A1: LSVQ introduces the concept of subtasks to improve the diversity of agents' policies. This technique is independent of the structures of the observation encoder or the value mixing network. LSVQ can be easily integrated with other methods such as QPLEX [1] and MAAC [2] and make further improvements based on their advanced modules. Therefore, we mainly compare LSVQ with other subtask-based methods in Section 1. A detailed related work section will be included in the future edition.
>
> Q2: General Dec-POMDPs have stochastic observations [3,4] and are sampled from a distribution instead of being projected via a deterministic function. The benchmark domains used in the paper are all special cases with deterministic observations (and even initial states) thus do not reflect the general characteristics of a general Dec-POMDP [4]. According to [3], the global state cannot be reconstructed from observations alone. The reviewer is not sure about the general validity of the proposed approach.
>
> A2: Although the observation function $\mathcal{O}$ can be stochastic, each agent would receive a sampled observation that could reflect the original distribution to a certain extent, and LSVQ can still work in such situations. Theoretically, the global state cannot be reconstructed perfectly from observations, but fortunately the global state is available during training. According to Figure 1, we assume the input of the subtask learner is the global state and consider the individual observations as vectors that have been processed by different channels. The global state is mapped into the latent space, then LSVQ attempts to reconstruct the global state based on the latents. The state reconstruction using latents is even harder than using observations alone, but as the quality of reconstruction improves with training, the latents become more informative as the mutual information $\mathcal{I}(s;\boldsymbol{z})$ increases. So the latents can serve as good subtask representations.
>
> Q3: The x-axes vary for different scenarios in Figures 3, 4, and 6, which is somewhat confusing. I wonder how the plots would look like, if the plots with only 2 million steps were run until 5 million steps as there could be a chance that the baselines outperform LSVQ in the long run.
>
> A3: For experiments on SMAC, most studies follow this setting: 5 million steps for the most difficult scenarios, and 2 million steps for others. For example, in *27m_vs_30m*, curves begin to rise at 500k steps, and 2 million steps are enough to predict the trends of the learning curves in practice. In *corridor*, curves rise from about 1.5 million steps, so 5 million steps is a proper setting. The learning curves would always stay the trend if the training continues.
>
> Q4: In the maps *5m_vs_6m*, *MMM2*, and *27m_vs_30m*, QMIX performs worse than reported in [5]. The reviewer suggests to evaluate on the newer SMACv2 [6], which exhibits more stochasticity and better aligns with the general Dec-POMDP setting.
>
> A4: Results on SMACv2 will be available in the appendix in the future edition.
>
> [1] S. Iqbal et al., "Actor-Attention-Critic for Multi-Agent Reinforcement Learning", ICML 2019
>
> [2] J. Wang et al., "QPLEX: Duplex Dueling Multi-Agent Q-Learning", ICLR 2021
>
> [3] F. Oliehoek et al., "A Concise Introduction to Decentralized POMDPs", 2016
>
> [4] X. Lyu et al., "A Deeper Understanding of State-Based Critics in Multi-Agent Reinforcement Learning", AAAI 2022
>
> [5] M. Samvelyan et al, "The StarCraft Multi-Agent Challenge", AAMAS 2019
>
> [6] B. Ellis et al., "SMACv2: An Improved Benchmark for Cooperative Multi-Agent Reinforcement Learning", 2022

---

> ### Comment · Reviewer_J7bq · 2023-11-21
> **Follow-Up**
>
> Thank you for the rebuttal.
>
> After reading the rebuttal and the other reviews, I find these shared concerns:
> - **Related work section:** Every scientific work needs a *context*, i.e., a discussion with related ideas to highlight conceptual differences, advantages, and disadvantages of the proposed idea. Simply claiming "independence" or "non-similarity" to prior work is not sufficient - there *has to be a context* to adequately assess novelty.
> - **Evaluation with SMACv2:** I would have appreciated some preliminary results, e.g., of a single scenario, as an exhaustive evaluation would have been infeasible within the short time of rebuttal and author discussion.
>
> I do not see any concrete revisions to the submission that have been promised in the rebuttal. This leaves me uncertain about the "future edition" regarding how the paper was rearranged, if it is consistent and self-contained, and if/how our suggestions have been considered. Therefore, I will not change my assessment.

---

> ### Author Response · Authors · 2023-11-21
>
> Thanks for your advice. We are still running the SMACv2 experiments now with limited computing resourses. As the rebuttal session ends at Nov 22nd, we will upload a global comment and a new version of our paper by that time, including the contents you mentioned.

---

### Official Review · Reviewer_qmnG · 2023-10-31

**Soundness:** 3 good
**Presentation:** 3 good
**Contribution:** 3 good
**Rating:** 3
**Confidence:** 5

**Summary:**

In this paper, the authors propose to Learn Subtask representations via Vector Quantization (LSVQ) for cooperative multi-agent reinforcement learning. The authors first design a novel subtask learner to reconstruct the global state through discrete subtask representations. And then, the subtask learner is integrated with other classic MARL frameworks to allocate subtasks to each agent. Experiments on SMAC and GRF have demonstrated the superior performance.

**Strengths:**

+ The paper is well-organized and well-written.
+ The proposed method can be easily integrated into various classic multi-agent reinforcement learning frameworks

**Weaknesses:**

+ The reviewer is concerned about the novelty of this paper. It seems that HSL(Heterogeneous Skill Learning) [1] shares the similar idea of this paper. A discussion on the difference between these two works should be added.

  [1] Liu, Yuntao, et al. "Heterogeneous Skill Learning for Multi-agent Tasks." *Advances in Neural Information Processing Systems* 35 (2022): 37011-37023.

+ The proposed method should be integrated into more advanced algorithms in MARL, such as MAPPO, HAPPO and MAT. More experiments should be conducted to strengthen the quality of this paper.

+ The performance on SMAC is saturated and the authors are suggested to supplement experimental results on SMAC v2.

+ To compare LSVQ-QMIX with QMIX, the scale of the Q network in QMIX is enlarged to achieve a comparable parameter count with that of LSVQ-QMIX. In the reviewer's opinion, the scale of the LSVQ-QMIX ought to be shrinked rather than enlarging the scale of QMIX.

+ In Figure 10, the results of LSVQ-QMIX against other baselines on GRF scenarios are not satisfactory compared with LDSA. Reasons for performance degradation should be analyzed and provided.

+ In the context of decentralization, how to assign subtasks to a specific agent is not only related to the observations, but also related to the overall situation. For example, there exist two identical agents which can both do task A and task B. Under the settings of decentralization, the two agents will do the same task, while we want them to do these two tasks, respectively.

**Questions:**

See weaknesses.

---

> ### Author Response · Authors · 2023-11-14
>
> Q1: The reviewer is concerned about the novelty of this paper. It seems that HSL shares the similar idea of this paper.
>
> A1: We have read the original paper of HSL and found that HSL is quite different from LSVQ but has a lot in common with other methods. HSL encodes the one-hot vector of skill-id $z_j$ to the latent representation space and introduces a learning objective to make the representations of different skill-ids distinguishable. This approach shares the same idea with LDSA [1], **with little global information in the representation**. The skill selecting process and the loss used to update the skill selector in HSL are also very similar to those in RODE [2]. The overall loss function includes several regularization terms, and **additional hyperparameters are needed for balancing the loss terms**. Instead, LSVQ integrates both the subtask encoding and subtask assignment into the VQ-VAE framework, which has a more solid theoretical foundation and naturally avoids generating identical representations for distinct subtasks. As described in Section 3.2 of our paper, the overall loss function is concise with few additional hyperparameters.
>
> Q2: The proposed method should be integrated into more advanced algorithms in MARL.
>
> A2: We integrated LSVQ with QMIX and MADDPG in our experiments to show the compatibility of LSVQ with both value decomposition methods and multi-agent actor-critic methods. There are two reasons for not using more advanced algorithms. First, as the baselines [1,2,3,4] use QMIX for value mixing, LSVQ keeps consistent with them for fairness. Second, the effectiveness of LSVQ is reflected in two aspects: whether it is superior to other subtask-based methods, and how much it improves the original algorithm (e.g., LSVQ-QMIX improves QMIX). Therefore, we think it is not necessary to integrate LSVQ with more advanced methods. But we agree that it would be better if we display the results of LSVQ-MAPPO instead of LSVQ-MADDPG in the ablation study.
>
> Q3: In the reviewer's opinion, the scale of the LSVQ-QMIX ought to be shrinked rather than enlarging the scale of QMIX for comparison.
>
> A3: We follow previous studies [1,3] which also enlarge the scale of QMIX for ablation studies. As shown in Figure 2(a) in our paper, the MLP, GRU, and the mixing network are exactly the same as those in QMIX. The design of the subtask learner follows the principle of simplicity, and further reduction of parameters can lead to insufficient network fitting capability. Therefore, we consider increasing the QMIX network for ablation to be reasonable.
>
> Q4: In Figure 10, the results of LSVQ-QMIX against other baselines on GRF scenarios are not satisfactory compared with LDSA. Reasons for performance degradation should be analyzed and provided.
>
> A4: Thank you for your careful reading of our paper. We apologize for not providing an analysis of this phenomenon. The most common reason is that different tasks require different capabilities of algorithms. For example, QMIX shows stable performances in many scenarios, but it fails in *corridor* which requires sufficient exploration. To remedy this, the exploration rate in QMIX needs to be increased [5]. The design of LDSA may be better suited to GRF scenarios, but it couldn't outperform LSVQ in overall performance. Further study on the GRF results would certainly help us to understand and improve LSVQ in greater depth.
>
> Q5: Subtask assignment is related to the overall situation as well as observation. For example, there exist two identical agents which can both do task A and task B. Under the settings of decentralization, the two agents will do the same task, while we want them to do these two tasks, respectively.
>
> A5: In our implementation, an agent's observation includes its agent-id, so the observations of two agents cannot be the same even if they are homogeneous. Follow the example in Q5, each agent can add its agent-id to its observation and the two agents can finally learn a joint policy to perform different subtasks based on their agent-ids. Therefore, two decentralized agents with identical external observations can still do different subtasks if necessary.
>
> [1] Yang, Mingyu, et al. "Ldsa: Learning dynamic subtask assignment in cooperative multi-agent reinforcement learning." Advances in Neural Information Processing Systems 35 (2022): 1698-1710.
>
> [2] Wang, Tonghan, et al. "Rode: Learning roles to decompose multi-agent tasks." arXiv preprint arXiv:2010.01523 (2020).
>
> [3] Wang, Tonghan, et al. "Roma: Multi-agent reinforcement learning with emergent roles." arXiv preprint arXiv:2003.08039 (2020).
>
> [4] Mahajan, Anuj, et al. "Maven: Multi-agent variational exploration." Advances in neural information processing systems 32 (2019).
>
> [5] Rashid, Tabish, et al. "Weighted qmix: Expanding monotonic value function factorisation for deep multi-agent reinforcement learning." Advances in neural information processing systems 33 (2020): 10199-10210.

---

### Official Review · Reviewer_FTem · 2023-11-02

**Soundness:** 3 good
**Presentation:** 4 excellent
**Contribution:** 3 good
**Rating:** 6
**Confidence:** 3

**Summary:**

The method presents a way to divide a task into sub-tasks using latent variable approach inspired by VQ-VAE, in which the learnt latent variables are used to reconstruct the global state. The learnt sub-task representations show a better decision making among agents and improves performance over different multi-agent tasks.

**Strengths:**

1. The document is well written and the motivation and discussion are clear.
2. The ablation studies address important questions about the method and are useful, especially the ablation on the number of sub-tasks.
3. The approach of using latent variables for sub-tasks is interesting direction for future work.

**Weaknesses:**

1. The method depends on finding the right number of sub-tasks K.

**Questions:**

1. Is it possible to study how entangled the latent representations and sub-tasks are? Do authors see a way in which disentanglement could help here?
2. How much tuning is needed and how difficult is it to find the right sub-task representation size for each method?

---

> ### Author Response · Authors · 2023-11-14
>
> Q1: Is it possible to study how entangled the latent representations and sub-tasks are? Do authors see a way in which disentanglement could help here?
>
> A1: This is an interesting perspective. We are not familiar with the area of disentanglement, so we took a brief look at it. The subtasks and representations are a one-to-one match, and it is more practical to study how entangled the latent representations and the global state are. In previous subtask-based methods [1,2], an additional regularization term is needed to make the representations of distinct subtasks distinguishable. Meanwhile, disentanglement helps to learn mutually independent subtask representations, which can replace the regularization term mentioned above and implicitly increase the discrepancy between representations with a more solid theoretical foundation. Existing disentanglement methods mainly focus on VAEs, and further study is required to combine disentanglement with VQ-VAE in our paper.
>
> Q2: How much tuning is needed and how difficult is it to find the right sub-task representation size for each method?
>
> A2: In our experiments, we found that LSVQ is very stable and insensitive to its hyperparameters. Theoretically, the number of subtasks $K$ corresponds to the number of distinct policies. A small $K$ would restrict the variety of the agents' behaviors, while a large $K$ leads to over-specialization of the strategy and unstable performances of agents. As shown in Appendix B.1, we set $K=8$ in all the scenarios across different environments and get promising results. Users can tune $K$ if necessary, but $K=8$ is a moderate choice and works well.
>
> [1] Yang, Mingyu, et al. "Ldsa: Learning dynamic subtask assignment in cooperative multi-agent reinforcement learning." Advances in Neural Information Processing Systems 35 (2022): 1698-1710.
>
> [2] Liu, Yuntao, et al. "Heterogeneous Skill Learning for Multi-agent Tasks." Advances in Neural Information Processing Systems 35 (2022): 37011-37023.

---

### Author Response · Authors · 2023-11-22

First, we would like to formally thank all the reviewers for the time and effort you have put into our paper. Your suggestions have guided us to improve the paper with a clearer target. We have already sent replies to each reviewer individually, and please remember to check them out.

We have now uploaded a new edition of the paper. Based on your suggestions, we've added a related work section in Appendix A, as well as the results on SMACv2 in Appendix D.2. The new experiments further confirm the effectiveness of LSVQ. We are very much looking forward to your comments and advice towards our modification to the paper, and we will keep refining our research.

---

### Meta-Review · Area_Chair_6EWC · 2023-12-08

**Metareview:**

This paper proposes Learn Subtask representations via Vector Quantization (LSVQ) for improving multi-agent RL. The key idea is to decompose tasks into sub-tasks, using a sub task learner. This learner is based on principles of vector quantization (akin to VQ-VAE). The learner's objective is to reconstruct global state through discrete sub task representations, enabling a more structure & efficient approach to task completion. This approach is demonstrated through various experiments on the starcraft multi-agent challenge, google research football environments, where this method is shown to be superior to existing methods. In general, this is an interesting research direction to enhance cooperative strategies and decision-making processes in complex multi-agent environments.

All reviewers agree that this is an interesting and important research topic in the context of MARL. However, there were concerns regarding contextualization to prior work, comparisons to baselines, questions about algorithmic choices, evaluating improvements and its significance. Like E766 suggested, it would be good to integrate LSVQ to broader MARL algorithms and there were unanswered questions on this front. Similarly, the SMACv2 experiments also needs more work and iterations as pointed by J7bq. The authors did run more experiments but the paper needs more revisions before it can be ready for acceptance.

I would encourage the authors to iterate on this paper one more time and resubmit it to a later conference.

**Justification For Why Not Higher Score:**

See meta review

**Justification For Why Not Lower Score:**

N/A

---

### Decision · Program_Chairs · 2024-01-16

Reject